# Organic–Inorganic Hybrid Device with a Novel Deep-Blue Emitter of a Donor–Acceptor Type, with ZnO Nanoparticles for Solution-Processed OLEDs

**DOI:** 10.3390/nano12213806

**Published:** 2022-10-28

**Authors:** Seokwoo Kang, Raveendra Jillella, Sunwoo Park, Sangshin Park, Joo Hwan Kim, Dakyeung Oh, Joonghan Kim, Jongwook Park

**Affiliations:** 1Integrated Engineering, Department of Chemical Engineering, Kyung Hee University, Yongin 17104, Korea; 2Department of Chemistry, The Catholic University of Korea, Bucheon 14662, Korea

**Keywords:** ZnO, blue emitter, solution process

## Abstract

Two new deep-blue emitters with bipolar properties based on an organoboron acceptor and carbazole donor were newly synthesized: 2-(9H-carbazol-9-yl)-5-(2,12-di-tert-butyl-5,9-dioxa-13b-boranaphtho [3,2,1-de]anthracen-7-yl)-5H-benzo[b]carbazole (TDBA-BCZ) and 5-(2,12-di-tert-butyl-5,9-dioxa-13b-boranaphtho [3,2,1-de]anthracen-7-yl)-8-phenyl-5,8-dihydroindolo[2,3-c]carbazole (TDBA-PCZ). The two emitters showed deep-blue and real-blue photoluminescence emission in their solution and film states, respectively. The doped spin-coated films were prepared using synthesized materials and showed a root-mean-square roughness of less than 0.52 nm, indicating excellent surface morphology. The doped devices, fabricated via a solution process using TDBA-BCZ and TDBA-PCZ as the dopants, showed electroluminescence peaks at 428 and 461 nm, corresponding to the Commission Internationale de L’éclairage (CIE) coordinates of (0.161, 0.046) and (0.151, 0.155), respectively. The external quantum efficiency (EQE)/current efficiency (CE) of the solution-processed forward devices, with TDBA-BCZ and TDBA-PCZ as dopants, were 7.73%/8.67 cd/A and 10.58%/14.24 cd/A, respectively. An inverted OLED device fabricated using rod-shaped ZnO nanoparticles as an electron injection layer showed a CE of 1.09 cd/A and an EQE of 0.30%.

## 1. Introduction

Organic light-emitting diodes (OLEDs) have received much attention in industry and academia because of their self-emission, fast response, flexibility, and full-color emission abilities [1,2,3,4,5,6,7,8,9,10,11,12,13,14,15,16,17,18,19,20,21,22]. Research on organoboron emitters containing an oxygen atom or nitrogen atom as new emitter candidates has been strongly focused on blue emitters. These materials use the multiple resonance (MR) effect, which leads to high-color-purity emissions with a narrow full-width at a half-maximum (FWHM) of ~20 nm and high oscillator strength, which, in turn, leads to a high photoluminescence quantum yield (PLQY). The MR effect is induced by localizing the orbital density of the highest occupied molecular orbital (HOMO) and the lowest unoccupied molecular orbital (LUMO) of each atom, such as carbon or boron. These molecular structures have been recently developed as blue emitters for high-performance full-color displays [23,24]. In particular, the National Television System Committee (NTSC) standards for large display panels, such as televisions, requires a color purity corresponding to a Commission Internationale de L’éclairage (CIE) color coordinate of y < 0.08 [25,26,27,28,29,30,31,32,33,34,35].

Most of the previously reported deep-blue OLED devices that contain an organoboron moiety have been prepared through a vacuum deposition process to achieve high performance. However, for the future mass production of large OLED displays, devices that can be prepared by a solution process such as spin-coating or inkjet printing are needed. Specifically, a solution-processed OLED small-molecule emitter that is amorphous and exhibits high solubility should be developed to facilitate simple processing [16,19]. Thus, small-molecule emitters are compared to conventional polymer-type emitters, which have the advantage of high performance in emitting devices and the disadvantage of difficult purification in material mass-production. For this reason, Jiang’s group developed a blue emitter fabricated by the solution processing of a small molecule, resulting in a device with an external quantum efficiency (EQE) of 13.4% [36]. Wang’s group has also achieved a maximum EQE (EQE_max_) of 10.5% using a small-molecule emitter that operates via the hot exciton mechanism and is prepared by a solution process [37].

In the present work, we take an alternative approach of focusing on the conventional forward OLED device performance. In the case of large-area television OLEDs, the driving circuit is based on an oxide thin-film transistor (TFT). This design is promising compared with that of OLEDs for mobile phones, in which the driving circuit is based on a TFT prepared by a low-temperature polysilicon (LTPS) method. An LTPS TFT is mainly driven by injecting hole carriers, whereas an oxide TFT is driven by injecting electron carriers. Accordingly, extensive research is being conducted on inverted OLED devices, along with research on conventional forward devices. In an inverted OLED device, the electrons are injected from indium tin oxide (ITO); in addition, ZnO, which has a suitable LUMO level of −4.3 eV and good electron transport performance, can be used as an electron injection layer (EIL). For example, Xu et al. reported inverted OLED device optimization using an electron-injecting layer of ZnO:CsCO_3_. The performance was controlled by work function and high electron carrier mobility. Jeon et al. also reported device characteristics using ZnO:PEI hybrid electron injection layers based on two key factors: molecular configuration and charge carrier properties [38,39]. In the present work, to investigate the characteristics of this EIL, we examined the nanoparticle structure of ZnO and the characteristics relevant to solution-processable OLEDs. Because a television requires a large screen, the devices must be fabricated using a solution process instead of the conventional evaporation method; we, therefore, prepared inverted devices using a solution process and evaluated their performance. We used a commercial ZnO nanoparticle solution to prepare a thin film of ZnO nanoparticles and investigated its optical properties and nanostructure. In order to realize high-efficiency luminous efficiency in a device fabricated by a solution process, we selected a cyclic organoboron moiety containing oxygen as an acceptor and a carbazole moiety as a donor and prepared two new emitters: 2-(9H-carbazol-9-yl)-5-(2,12-di-tert-butyl-5,9-dioxa-13b-boranaphtho [3,2,1-de]anthracen-7-yl)-5H-benzo[b]carbazole (TDBA-BCZ) and 5-(2,12-di-tert-butyl-5,9-dioxa-13b-boranaphtho [3,2,1-de]anthracen-7-yl)-8-phenyl-5,8-dihydroindolo [2,3-c]carbazole (TDBA-PCZ).

TDBA-BCZ and TDBA-PCZ contain an organoboron chemical structure with high planarity, and excellent surface morphology can be expected after the solution process. The two new emitters have a highly twisted structure and are expected to exhibit high solubility because of the large dihedral angle inside the molecule. In this paper, we report the optical properties and the results of the density functional theory (DFT) calculations of the two emitters, as well as the electroluminescence (EL) performance of a doped device with an inverted device configuration prepared using a solution process.

## 2. Experimental

The synthesis methods of TDBA, **1**, **2**, **3**, and **4**, which are used in the present work as intermediate compounds (Appendix A), have been reported elsewhere [24,40]. The ^1^H and ^13^C nuclear magnetic resonance (NMR) data are presented in Appendix A. 

### 2.1. Synthesis of 2-(9H-Carbazol-9-yl)-5-(2,12-di-tert-butyl-5,9-dioxa-13b-boranaphtho [3,2,1-de]anthracen-7-yl)-5H-benzo[b]carbazole (TDBA-BCZ)

A mixture containing 2-(9H-carbazol-9-yl)-5H-benzo[b]carbazole (4) (0.5 g, 1.3 mmol), 7-bromo-2,12-di-tert-butyl-5,9-dioxa-13b-boranaphtho [3,2,1-de]anthracene (TDBA) (0.72 g, 1.56 mmol), sodium tert-butoxide (187.3 mg, 1.95 mmol), tri-tert-butylphosphine/HBF4 (22.8 mg, 0.075 mmol), and tris(dibenzylideneacetone)dipalladium(0) (Pd_2_(dba)_3_) (32.04 mg, 0.035 mmol) was dissolved in toluene (20 mL). The resultant solution was purged with N2 and then stirred at 120 °C for 18 h in the presence of N_2_. The reaction mixture was then cooled to room temperature and diluted with ethyl acetate before being filtered through silica gel and concentrated. The residue was purified by column chromatography and recrystallized from methanol, giving TDBA-BCZ (643 mg, yield 65%) as a white solid. ^1^H-NMR (400 MHz, CDCl_3_): δ 8.84 (s, 2H), 8.35 (s, 1H), 8.20–8.17 (m, 3H), 8.01 (d, J = 8.0 Hz, 1H), 7.84 (dd, J = 8.0 Hz, 6.0 Hz, 2H), 7.78 (d, J = 8 Hz, 1H), 7.57–7.51 (m, 5H), 7.48 (s, 2H), 7.45–7.39 (m, 5H), 7.31–7.27 (m, 2H), 7.22–7.18 (m, 1H), 1.29 (s, 18H) ppm; ^13^C-NMR (125 MHz, CDCl_3_): 158.8, 158.7, 145.7, 145.3, 141.9, 141.0, 136.4, 133.8, 132.0, 130.8, 130.4, 129.3, 125.9, 125.6, 125.3, 124.9, 124.8, 123.2, 122.6, 122.2, 122.0, 120.4, 119.7, 119.0, 118.8, 118.2, 111.8, 110.0, 109.1, 34.7, 31.6 ppm. High-resolution mass spectrometry (HRMS) (fast atom bombardment mass spectrometry (FAB-MS), m/z): calcd. for C_54_H_43_BN_2_O_2_, 762.34; found: 763.3490 [M]+.

### 2.2. Synthesis of 4-Bromo-9-phenyl-9H-carbazole (**5**)

A mixture of 4-bromo-9H-carbazole (10 g, 0.041 mol), potassium phosphate (28.28 g, 0.121 mol), and CuI (3.87 g, 0.020 mol) was added to 20 mL of anhydrous 1,4-dioxane solution in a three-neck round-bottom flask under a N_2_ atmosphere. After the mixture was heated to 80 °C, iodobenzene (12.43 g, 0.051 mol) and ethylenediamine (3.6 gm, 0.061 mol) were added using a 3 mL syringe. The reaction mixture was stirred and refluxed for 16 h under a N_2_ atmosphere and was then diluted with dichloromethane and washed with water. The organic layer was dried with anhydrous MgSO_4_, and the solvent was removed under reduced pressure. The crude product was purified by column chromatography on silica gel using dichloromethane/n-hexane to give 5 (12.18 g, yield 85%). ^1^H-NMR (300 MHz, CDCl_3_): δ 8.84 (d, J = 8.0 Hz, 1H), 7.65–7.23 (m, 11H) ppm.

### 2.3. Synthesis of 9-Phenyl-4-(4,4,5,5-tetramethyl-1,3,2-dioxaborolan-2-yl)-9H-carbazole (**6**)

A mixture of 5 (6.1 g, 19.0 mmol), bis(pinacolato)diboron (7.3 g, 28 mmol), and potassium acetate (7.66 g, 76 mmol) in 100 mL of dry toluene was bubbled with N_2_ for 15 min. 1,1′-Bis(diphenylphosphino)ferrocene]dichloropalladium(II) (Pd(dppf)Cl_2_, 0.69 g, 0.95 mmol) was then added. The mixture was heated at 100 °C (oil bath) for 18 h. After cooling, the mixture was filtered through Celite. Water was then added, and the organic phase was extracted with ethyl acetate three times (30 mL × 3). The organic layer was washed with water (10 mL × 3) and brine (50 mL × 1), dried over MgSO_4_, filtered, and concentrated. The residue was purified by silica-gel column chromatography (eluent: hexane) to give 6 as a pale-yellow solid (4.8 g, yield 87.2%). ^1^H-NMR (400 MHz, CDCl_3_) δ 9.09 (dt, J = 8.0, 1.0 Hz, 1H), 7.80 (dd, J = 7.1, 1.2 Hz, 1H), 7.63–7.56 (m, 2H), 7.54–7.45 (m, 4H), 7.42–7.36 (m, 2H), 7.34 (ddd, J = 8.2, 1.4, 0.7 Hz, 1H), 7.28 (ddd, J = 8.1, 6.8, 1.3 Hz, 1H), 1.49 (s, 12H) ppm.

### 2.4. Synthesis of 4-(2-Nitrophenyl)-9-phenyl-9H-carbazole (**7**)

Compound **6** (4.8 g, 13.0 mmol), 2-bromo nitrobenzene (5.35 g, 26.0 mmol), and tetrakis(triphenylphosphine)palladium(0) (0.75 g, 0.6 mmol) were dissolved in 40 mL of dry THF; A total of 50 mL of an aqueous solution of K_2_CO_3_ (5.50 g, 39.0 mmol) was then added. After the solution was degassed by Ar bubbling, the reaction mixture was heated to 110 °C for 24 h and then cooled to room temperature. The reaction mixture was then extracted with ethyl acetate, and all the organic layers were combined and subsequently concentrated under reduced pressure. The obtained crude product was purified by column chromatography with 1:10 (ethyl acetate:n-hexane) to obtain 7 (3.8 g, yield 80.2%). ^1^H-NMR (400 MHz, CDCl_3_): δ 8.35(d, J = 8.0 Hz, 1H), 7.65–7.56 (m, 6H), 6.85–6.21 (m, 3H) ppm.

### 2.5. Synthesis of 5-Phenyl-5,8-dihydroindolo [2,3-c]carbazole (**8**)

A mixture of 7 (1.5 g, 4.1 mmol) and triphenylphosphine (5.59 g, 21.1 mmol) in 1,2-dichlorobenzene (12 mL) was added to a flask, and the reaction mixture was stirred and refluxed for 12 h under a N_2_ atmosphere. The reaction mixture was then cooled to room temperature; the 1,2-dichlorobenzene solvent was removed by filtration through a column with hexane as the eluent, and the remaining crude mixture was filtered through ethyl acetate. The crude product was purified by column chromatography on silica gel using ethyl acetate/n-hexane as the eluent. The product was obtained as a brown powder (0.7 g, yield 52%). ^1^H-NMR (400 MHz, DMSO) δ 11.56 (s, 1H), 8.86–8.73 (m, 2H), 7.72–7.61 (m, 6H), 7.60–7.50 (m, 1H), 7.48–7.39 (m, 5H), 7.32 (ddd, J = 8.2, 7.0, 1.2 Hz, 1H).

### 2.6. Synthesis of 5-(2,12-Di-tert-butyl-5,9-dioxa-13b-boranaphtho [3,2,1-de]anthracen-7-yl)-8-phenyl-5,8-dihydroindolo [2,3-c]carbazole (TDBA-PCZ)

A mixture of 8 (200 mg, 0.6 mmol), 7-bromo-2,12-di-tert-butyl-5,9-dioxa-13b-boranaphtho [3,2,1-de]anthracene (305.0 mg, 0.7 mmol), sodium tert-butoxide (96 mg, 0.9 mmol), tri-tert-butylphosphine (0.12 mg, 0.1 mmol), and Pd_2_(dba)_3_ (56.0 mg, 0.1 mmol) was dissolved in toluene (6 mL). After the mixture was purged with N2, it was stirred at 120 °C for 18 h in the presence of N_2_. The reaction mixture was then cooled to room temperature and diluted with dichloromethane. The mixture was filtered through silica gel and concentrated. The residue was purified by column chromatography and recrystallized from methanol give TDBA-PCZ (247.7 mg, yield 58%) as a yellow solid. ^1^H-NMR (400 MHz, CDCl_3_) δ 8.99 (td, J = 6.3, 5.9, 2.2 Hz, 2H), 8.79 (d, J = 2.5 Hz, 2H), 7.83–7.72 (m, 4H), 7.64–7.62 (m, 3H), 7.56–7.44 (m, 12H), 1.51 (s, 18H) ppm; ^13^C-NMR (125 MHz, CDCl_3_): 158.8, 158.6, 145.4, 143.7, 141.2, 137.2, 136.3, 131.8, 130.4, 130.0, 128.0, 127.8, 125.6, 123.6, 120.3, 119.7, 118.1, 117.6, 110.4, 110.0, 109.3, 109.1, 107.4, 34.7, 31.6. HRMS (FAB-MS, m/z): calcd. for C_50_H_41_BN_2_O_2_, 712.33; found: 712.3270 [M]+.

## 3. Results and Discussion

### 3.1. Molecular Design, Synthesis, and Characterization of Organic Blue Emitters

For the molecular design of the new deep-blue emitters with a bipolar character, TDBA with a tert-butyl group substituted at the terminal group of an organoboron moiety containing boron and oxygen atoms was used as an acceptor because TDBA has recently been studied for its excellent photoelectric properties [23,24]. TDBA exhibits high structural rigidity, enabling it to provide a low energy loss of excitons, a high photoluminescence quantum yield (PLQY), and excellent color purity because of the narrow full-width at half-maximum (FWHM) characteristics of its emission. In addition, the introduced tert-butyl group can prevent molecular packing between the molecules and preserve the optical properties of the original chemical structure. As a donor group for the bipolar emitters, we designed two new donor groups based on carbazole. The first group is a benzocarbazole (BCZ) moiety that is a derivative with a phenyl ring fused to carbazole (CZ); it is positioned at the para site relative to the boron atom of TDBA. BCZ has a planar structure and can also provide a smooth surface, making it compatible with solution processing [40]. The CZ introduced at the para position of BCZ can maintain blue emission by suppressing intermolecular interactions while maintaining the high hole mobility of the compound. The second compound uses a phenyl indolocarbazole group (PCZ), in which two CZs are fused. Because it contains two nitrogen atoms in a molecule, it demonstrates excellent donating ability and can also form dense films as a result of increased planarity. After synthesizing the two donor moieties based on the CZ moiety, we successfully prepared the new bipolar materials TDBA-BCZ and TDBA-PCZ. The synthesis route and molecular structures of TDBA-BCZ and TDBA-PCZ are shown in Figure 1 and Appendix A, respectively. TDBA-BCZ was synthesized through nitration, nucleophilic substitution, Suzuki coupling reaction, Cadogan cyclization, and Buchwald–Hartwig amination. TDBA-PCZ was synthesized by Ullmann reaction, Pd-catalyzed borylation, Suzuki coupling reaction, Cadogan cyclization, and Buchwald–Hartwig amination. All compounds were purified by recrystallization and column chromatography. The synthesized compounds were characterized using NMR spectroscopy, mass spectrometry, and elemental analysis.

### 3.2. Photophysical Properties of Organic Blue Emitters

Figure 2a,b show the UV–Visible (UV–Vis) absorption spectra and photoluminescence (PL) spectra of the newly synthesized materials in the solution and film states, respectively; their detailed photophysical properties are summarized in Table 1.

The UV–Vis absorption spectra of TDBA-BCZ and TDBA-PCZ in the solution state show similar spectral shapes. The absorption in the wavelength range below 350 nm is attributed to the π–π * and n–π * absorption from the donor moieties, and the absorption peak at ~380 nm arises from the intramolecular charge transfer (ICT) transition between the donor and acceptor moieties. The maximum photoluminescence (PL_max_) values of TDBA-BCZ and TDBA-PCZ are 413 and 451 nm, which are in the deep-blue region. The emission of TDBA-PCZ is red-shifted when compared with that of TDBA-BCZ because of the difference in its donating ability. This behavior is explained by the transition dipole moments of TDBA-BCZ and TDBA-PCZ, obtained via time-dependent (TD) density functional theory (DFT) calculations: 27.5 and 29.9 debye, respectively. In addition, the FWHM values of the two emitters are 55 and 60 nm, which are related to the typical ICT of the donor–acceptor emitters. The UV–Vis absorption spectra of both emitters in the spin-coated film state were similar to the spectra of the corresponding emitters in the solution state. In addition, the PL_max_ values of TDBA-BCZ and TDBA-PCZ in the film state were 436 and 470 nm, respectively. Thus, the emissions of TDBA-BCZ and TDBA-PCZ in the film state were red-shifted by 23 and 19 nm, respectively, compared with those of TDBA-BCZ and TDBA-PCZ in the solution state; this shift is attributed to the packing of the molecules in the film state. In the diluted solution state, there is enough molecular distance, and this does not create an additional molecular interaction in the excited state. According to the decreased molecular distance in the film state, the increased molecular interaction from molecular packing causes energy loss and red-shifted photoluminescence emission. The FWHM values for TDBA-BCZ and TDBA-PCZ in the film state were 58 and 61 nm, respectively, or approximately 1–3 nm broader than those for TDBA-BCZ and TDBA-PCZ in the solution state. This small broadening of 1–3 nm is attributed to the twisted structure with a dihedral angle greater than ~52° between the donor and acceptor groups of TDBA-BCZ and TDBA-PCZ (Appendix A).

To confirm the bipolar characteristics of the newly synthesized emitters, we conducted solvatochromic experiments using several solvents with different polarities (Appendix A). As the polarity of the solvent increased, TDBA-BCZ and TDBA-PCZ, which have bipolar characteristics, were confirmed to exhibit red-shifted emissions because of strong ICT characteristics. The difference between the S1 and T1 energy levels (Δ*E*_ST_) of TDBA-BCZ and TDBA-PCZ were measured from their PL spectra recorded at room temperature (RT) and at a low temperature (LT) of 77 K (Appendix A and Table 1). The Δ*E*_ST_ values of TDBA-BCZ and TDBA-PCZ were 0.32 and 0.13 eV, respectively. The absolute PLQY values of TDBA-BCZ and TDBA-PCZ in the solution, neat-film, and doped-film states were measured (Table 1). The corresponding absolute PLQY values were 34%, 36%, and 37%, respectively, for TDBA-BCZ and 48%, 58%, and 52%, respectively, for TDBA-PCZ. Interestingly, both emitters exhibited higher PLQY values in the neat-film state than in the solution state, suggesting an aggregation-induced emission (AIE) effect [41,42,43,44]. In order to investigate the AIE effect of the two emitters, we measured the PLQY values of TDBA-BCZ and TDBA-PCZ in THF/water mixtures with different water fractions (fws) (Appendix A). For the two emitters, the PLQY value increased with an increasing water fraction ratio. This phenomenon is attributed to the AIE phenomenon without the conventional AIE chemical structure and is caused by the highly twisted structure. It is also attributable to the nonradiative process being inhibited by the limitation of intermolecular motion [41]. We also recorded time-resolved PL spectra to examine the excited-state behaviors of the two emitters (Figure 2c,d). The fluorescence lifetimes of TDBA-BCZ and TDBA-PCZ were 3.13 and 5.07 ns, respectively. In addition, these materials showed lifetimes of 7.99 and 12.2 ns, respectively, in the doped-film state, which means that the doped films have longer lifetimes than the neat films because of energy transfer from the host to the dopant.

### 3.3. Surface Morphology and Thermal Properties of the Organic Blue Emitters

The morphology of the coated emitter film surface is important when fabricating solution-processed OLED devices. A defect-free uniform surface condition prevents overcurrent and leakage current during OLED device operation, increases the stability of the light-emitting layer, and finally leads to normal device performance. Atomic force microscopy (AFM) and confocal microscopy were carried out to confirm the surface properties of the spin-coated film using the synthesized materials (Figure 3). Figure 3a,b show the AFM images and confocal images of the films doped with TDBA-BCZ and TDBA-PCZ in an mCP host.

The root-mean-square (RMS) roughness values of the doped TDBA-BCZ and TDBA-PCZ films are 0.515 and 0.386 nm, respectively, which indicates that the films exhibit smooth surfaces and are uniform. We also confirmed, through confocal images, that the surface properties are very good. However, in the case of the nondoped films, RMS values of 0.660 and 0.404 nm were obtained for TDBA-BCZ and TDBA-PCZ, respectively, indicating slightly rough surfaces (Figure 3c,d). In addition, slight surface imperfections were confirmed through the confocal images. On the basis of these results, we presumed that a doped OLED device fabricated using the prepared emitters would show good EL performance with good interfacial properties.

The thermal properties of the newly synthesized materials were evaluated by thermogravimetric analysis (TGA) and differential scanning calorimetry (DSC) (Appendix A). TDBA-BCZ and TDBA-PCZ showed high thermal stability compared with conventional emitters, with decomposition temperatures (*T_d_*) of 458 and 452 °C and glass-transition temperatures (*T_g_*) of 186 and 171 °C, respectively. TDBA-BCZ exhibits relatively better thermal stability than TDBA-PCZ, possibly because of its higher molecular weight. Emitters with high thermal stability generally exhibit better device stability in terms of morphology changes during the operation of an OLED device.

### 3.4. Characterization and Photophysical Properties of ZnO Nanoparticles

To evaluate the suitability of ZnO nanoparticles for use in the EIL of an inverted OLED device, we recorded their X-ray diffraction (XRD) pattern to characterize their crystal lattice structure. We further characterized the ZnO nanoparticles using UV–Vis absorption spectroscopy, PL spectroscopy, scanning electron microscopy (SEM), atomic force microscopy (AFM), and transmission electron microscopy (TEM) to confirm their optical properties, surface characteristics, and crystal size, which are related to device performance (Figure 4 and Figure 5). Figure 4a shows a typical XRD pattern of ZnO nanoparticles in the 2θ range from 30° to 70°. The reflection peaks appear in the order (100), (002), (101), (102), (110), (103), and (112), and a wurtzite-type structure is confirmed [45]. In particular, the (002) reflection peak at 34.4° is located between the (100) reflection at 31.8° and the (101) reflection at 36.2°. The relatively sharp peak of the (002) reflection is a rod-type domain, as reported by the groups of Wilken and Pu [45,46]. The sharp reflections observed in the present study are attributed to rod-like shapes, consistent with the TEM observations and the band-edge value obtained from the UV–Vis spectrum.

The UV–Vis absorption spectrum of the ZnO spin-coated thin film showed an absorption maximum at 356 nm (Figure 4b), and a band-edge value of 3.30 eV was obtained when the optical energy gap was calculated. We derived the optical bandgap (i.e., the band-edge value) by determining the absorption edges from the plots of (αhν)^2^ vs. (hν), where α, h, and ν are the absorbance, Planck’s constant, and the frequency of light, respectively. This value is the same as that of a rod-like ZnO film, as reported by the groups of Clarke and Pu [46,47]. As the shape of the crystals in a ZnO nanoparticle change from spherical to rod-like, the bandgap decreases, and the absorption edge is red-shifted. Similar effects can be confirmed through the PL spectrum (Figure 4b). The emission peaks of ZnO at 350 and 385 nm were confirmed, and the peak at 550 nm, associated with the trap state of typical ZnO spherical particles, was not observed. The trap-state peak of 550 nm in the PL spectrum reported by Meijerink’s group is attributable to the formation of heterogeneous interfaces around the ZnO nanoparticles [48]. In the case of spherical particles with a small nanoparticle size, the trap-site luminescence at 550 nm can provide numerous heterogeneous interfaces around the nanoparticle. However, in the present work, a luminescence at 550 nm is not observed, similar to the case of the rod-shaped particles reported by Pu’s group. The rod-like ZnO nanoparticle film prepared in the present study is helpful for improving device efficiency because of its fast electron carrier performance and a small number of heterogeneous interfaces. In addition, the ZnO layer was confirmed to exhibit no absorption of visible light in the wavelength range from 400 to 800 nm in its UV–Vis absorption spectrum (Figure 4b); thus, we confirmed that it could be freely used in forward conventional devices and inverted devices.

A ZnO nanoparticle film formed on an ITO should exhibit a smooth surface to reduce the leakage current that can occur from a rough surface. In addition, it is advantageous for a ZnO film to have a high density to reduce heterogeneous interfaces or defects within the film. The SEM and AFM images of a ZnO film spin-coated onto an ITO substrate (Figure 5a,b) confirm that a high-density and homogeneous ZnO film layer was formed on the ITO. In particular, the average roughness of the solution-processed films, which is an important parameter for films used in OLED devices, was observed to be 2.09 nm by AFM, confirming that it can be normally used in an OLED device. The TEM image in Figure 5c confirms the formation of ZnO nanoparticles with a rod-like shape and an average size of 10 nm × 20 nm. We found that the ZnO used in the present work is suitable for solution processing and that the morphology of the resultant thin films is excellent. These features should result in the improved performance of those devices fabricated using the ZnO nanoparticles.

### 3.5. Electroluminescence Properties of Conventional and Inverted OLED Devices

To confirm the EL performance of TDBA-BCZ and TDBA-PCZ, the conventional forward-driving doped OLED devices were fabricated using a hybrid solution–evaporation process. The device structure was ITO (anode, 150 nm)/poly(3,4-ethylenedioxythiophene):poly(styrenesulfonate) (PEDOT:PSS) (solution process, 40 nm)/poly(9-vinylcarbazole) (PVK) (solution process, 20 nm)/1,3-bis(N-carbazolyl)benzene (mCP): 30% dopant (TDBA-BCZ or TDBA-PCZ) (solution process)/1,3,5-tris(1-phenyl-1H-benzimidazol-2-yl)benzene (TPBi) (evaporation, 40 nm)/LiF (evaporation, 1 nm)/Al (cathode, 200 nm). PEDOT:PSS and PVK were used as the hole injection layer (HIL) and the hole transport layer (HTL), respectively. mCP was used as the host material, and the two synthesized emitters (TDBA-BCZ and TDBA-PCZ) were used as a dopant material in the host–dopant system. A TPBi film was used as an electron transporting and hole blocking layer.

The current density–voltage–luminance (J–V–L) graphs of the OLED devices fabricated using the two dopant emitters show typical diode characteristics (Figure 6c). The EL performance of the solution-processed OLED devices is shown in Figure 6. Figure 6a,b show the energy-level diagrams of the conventional and inverted device structures, respectively, and Figure 6c,f show the EQE and power efficiency according to the current density and EL spectra for the solution-processed OLED devices, respectively. The related experimental results are summarized in Table 2.

The turn-on voltage (*V*_on_) of the devices with TDBA-BCZ and TDBA-PCZ as dopants was 4.89 and 4.29 eV, respectively. The solution-processed OLED devices fabricated using TDBA-BCZ and TDBA-PCZ as the dopants demonstrated excellent EL performance, including CE_max_ values of 8.67 cd/A and 14.24 cd/A and EQE_max_ values of 7.73% and 10.14%, respectively. Both devices showed a roll-off efficiency of ~5% loss, even at a high brightness of 1000 cd/m^2^. This performance can be explained by the smooth charge injection and the optimized device structure that originate from the bipolar character of the newly synthesized molecular structures. In addition, because the light-emitting layer showed an excellent RMS of 0.515 and 0.386 nm in the spin-coated film state, it formed a smooth surface and effectively prevented the trapping of charge carriers during device operation. As shown in Figure 6d, for TDBA-BCZ and TDBA-PCZ, the maximum EL wavelengths are 428 nm (deep-blue region) and 461 nm (blue region), and the CIE color coordinates are (0.161, 0.046) and (0.151, 0.155), respectively. The FWHM values of TDBA-BCZ and TDBA-PCZ are relatively small (47 and 58 nm, respectively), providing good color purity. This is explained by the simplified electron transition, which comes from the decreased molecular packing as well as the twisted chemical structure. Notably, TDBA-BCZ shows an excellent EQE of 7.73% in the deep-blue region of CIE *y* = 0.046; this EQE is the highest value reported for a solution-processed OLED device that also satisfies the ultra-high-density television (UHD-TV) specification of CIE *y* < 0.05. Additionally, the TDBA-PCZ device showed very high efficiency, an EQE of 10.14% in CIE y of 0.155, and real blue color. This means that the device can be used for a mobile display. In order to evaluate the luminescence property of the OLED emitters according to the excitation energy density, the EL spectra were measured according to the voltage variation. As shown in Appendix A, the EL spectrum of the solution processed OLED device does not change as the voltage increases.

Single-carrier devices were fabricated to evaluate the charge transport ability of TDBA-BCZ and TDBA-PCZ (Appendix A). The current density of hole-only devices (HODs) and electron-only devices (EODs) fabricated using the two emitters was measured as the voltage was increased. In the TDBA-BCZ-based devices, the hole and electron injection rates were slightly different. By contrast, in the TDBA-PCZ-base devices, the hole and electron injection rates were confirmed to be approximately the same. Emitters, such as TDBA-PCZ, with such bipolar characteristics can realize high EL performance because the charge injection is smooth and can induce a broad recombination zone. In addition, to evaluate the stability of the fabricated OLED devices, device lifetime was measured at 1000 cd/m^2^. The operational lifetime (LT_50_) means the device lifetime, after which the final light intensity reaches 50% of its initial intensity. The LT_50_ values of TDBA-BCZ and TDBA-PCZ were 15.4 and 21.5 h, as shown in Appendix A. In solution-processed OLED device applications, the device lifetime is relatively excellent compared to those within previously reported papers [49,50].

The inverted OLED device configuration and solution-processed OLEDs have recently been attracting attention in the research field of large displays, such as televisions. Thus, by using TDBA-PCZ, which exhibits high efficiency among the two synthesized materials, a hybrid-type inverted device prepared using the solution process was also fabricated. ZnO nanoparticles have an average diameter of 23.5 nm and a polydispersity index of 0.15, as shown in Appendix A. The structure of the inverted device was ITO (cathode)/ZnO (solution, 30 nm)/polyethyleneimine (PEI) (solution, 5 nm)/mCP:30 wt% TDBA-PCZ/1,1-Bis[(di-4-tolylamino)phenyl]cyclohexane (TAPC) (evaporation, 20 nm)/MoO_3_ (evaporation, 10 nm)/Al (anode). The band diagram of the layers is shown in Figure 6b. ZnO was used as the EIL, and PEI was used as the ETL and the interlayer. PEI can improve the mismatched interfacial properties between the inorganic material ZnO and the organic emitting (EML) layer [38,39]. TAPC and MoO_3_ were used as the HTL and HIL. Table 2 summarizes the efficiency of the fabricated inverted OLED device. The device exhibited a luminance efficiency of 1.09 cd/A and an EQE of 0.30%. It showed overall lower efficiency than the forward device, which is attributed to the relatively large energy barrier of the holes and electrons in the device configuration. Therefore, in the future, the charge balance can be improved through the selection of a different carrier transport layer and the development of additional thickness controls; in addition, the efficiency can be improved by lowering the hole and electron injection barriers using a doped-state ETL and HTL. Further studies on the replacement of new materials instead of PEI to decrease the LUMO level are underway. Additionally, the efficiency of the device can be improved by decreasing the pin hole number of the ZnO layer. We will separately report on this in the future.

Thus far, only a few papers on inverted solution-processed OLEDs fabricated using small molecules have been reported [51,52,53]. In the present study, the inverted blue OLED device, fabricated using small molecules and a solution process, was demonstrated. In conclusion, both the forward and inverted devices based on a hybrid-type solution–evaporation process can be fabricated by developing a new blue dopant material compatible with solution processes and by using rod-shaped ZnO nanoparticles. We expect that the production of large-area OLEDs will be possible in the future by applying the all-solution process.

## 4. Conclusions

New bipolar deep-blue-light-emitting materials TDBA-BCZ and TDBA-PCZ, which are based on an organoboron moiety, including an oxygen atom as an acceptor and an optimized carbazole moiety as a donor group, were designed and synthesized. TDBA-BCZ and TDBA-PCZ were confirmed to emit photoluminescence in the deep-blue and real-blue regions of 413 nm and 451 nm in the solution state and to exhibit bipolar characteristics under various solvent conditions. Both materials exhibit Δ*E*_ST_ values smaller than 1.0 eV, as determined from actual experimental values and theoretical calculations. The two emitters showed an RMS value of 0.52 nm or less in the film state after the spin-coating process, confirming that they exhibit excellent surface morphology. The solution-processed OLED devices, fabricated using TDBA-BCZ and TDBA-PCZ as dopants, showed a high current efficiency of 8.67 and 14.24 cd/A, and EQE values of 7.73 and 10.14%, respectively. In particular, TDBA-BCZ showed an excellent EL performance, with a CIE *y* value of 0.046, which is lower than the blue CIE *y* value of 0.08 specified in the NTSC standard. This result is one of the highest EL performances reported for a deep-blue-emitting OLED device fabricated using a solution process. The efficiency of the inverted OLED device, fabricated using rod-shaped ZnO nanoparticles as an EIL material, exhibited a luminance efficiency of 1.09 cd/A and an EQE of 0.30%. We speculate that these results will lead to material enhancement in the performance of solution-processed OLEDs for televisions.

## Figures and Tables

**Figure 1 nanomaterials-12-03806-f001:**
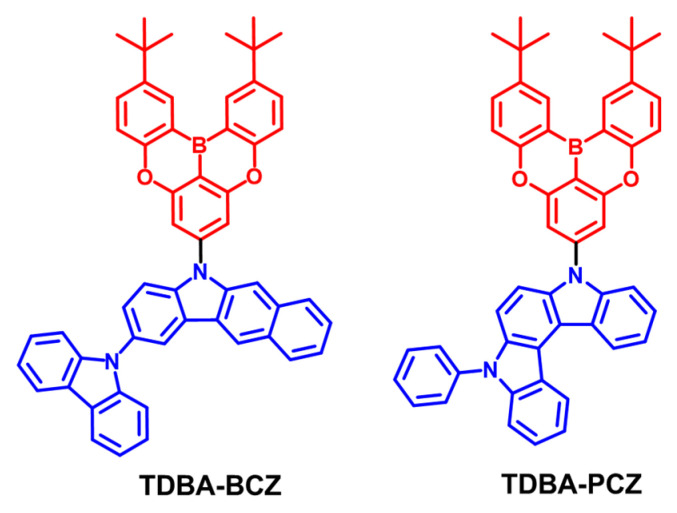
Molecular structures of the synthesized emitters with bipolar properties: TDBA-BCZ and TDBA-PCZ.

**Figure 2 nanomaterials-12-03806-f002:**
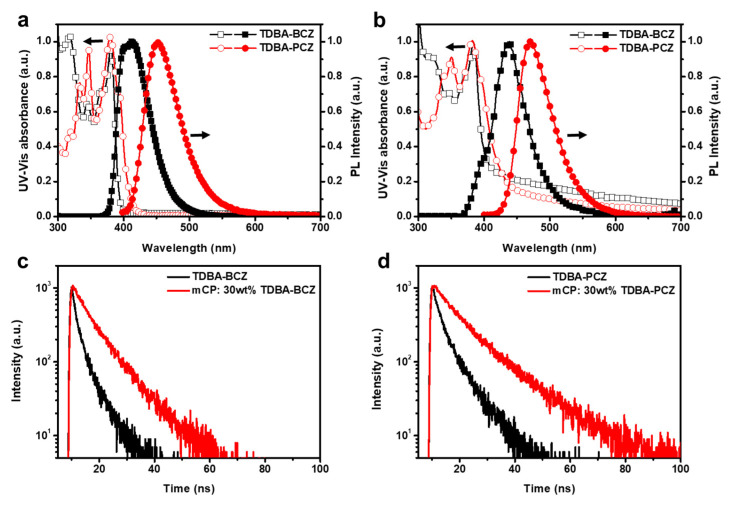
UV–Vis absorption and photoluminescence (PL) spectra of TDBA-BCZ and TDBA-PCZ in the (**a**) solution and (**b**) spin-coated film states (nondoped). (**c**) Transient PL decay curves of (**c**) TDBA-BCZ and (**d**) TDBA-PCZ in the spin-coated film state (nondoped and 30 wt% doped 1,3-bis(N-carbazolyl)benzene (mCP)).

**Figure 3 nanomaterials-12-03806-f003:**
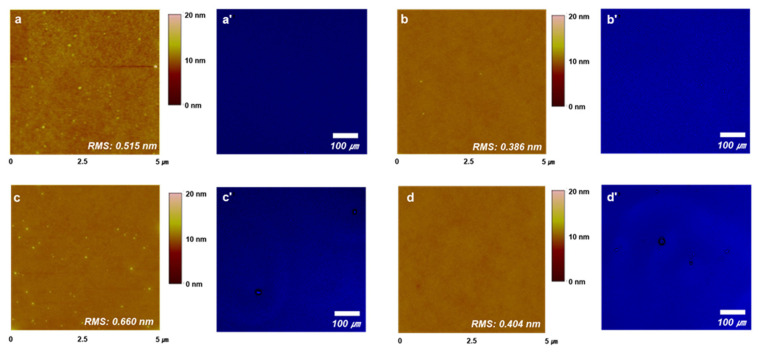
The AFM and confocal images of (**a**,**a′**,**c**,**c′**) TDBA-BCZ and (**b**,**b′**,**d**,**d′**) TDBA-PCZ in the doped (**top row**) and nondoped (**bottom row**)-film states.

**Figure 4 nanomaterials-12-03806-f004:**
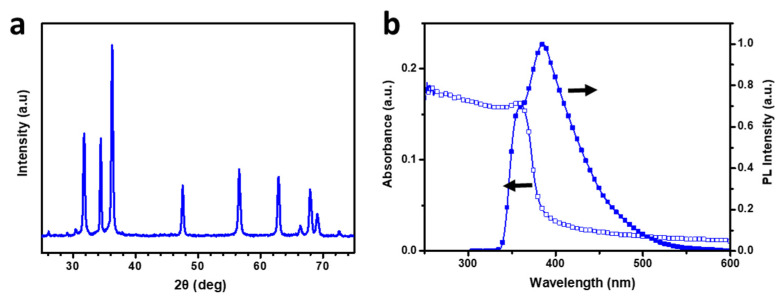
(**a**) XRD pattern and (**b**) UV–Vis absorption and PL spectra of a ZnO layer.

**Figure 5 nanomaterials-12-03806-f005:**
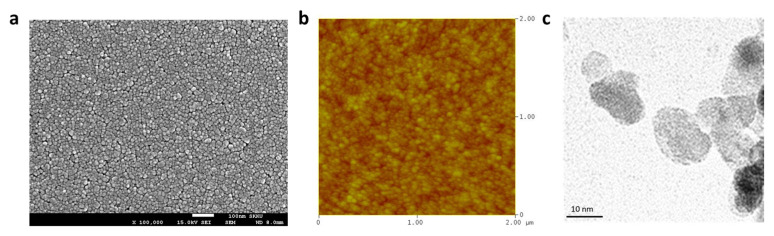
(**a**) SEM image of a ZnO layer on ITO, (**b**) AFM image of a ZnO layer on ITO, and (**c**) TEM image of ZnO nanoparticles.

**Figure 6 nanomaterials-12-03806-f006:**
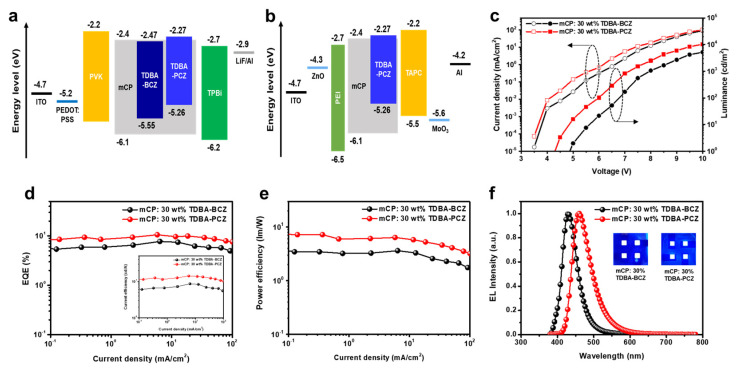
(**a**) Energy−level diagrams of a conventional forward device, (**b**) energy−level diagrams of an inverted device, (**c**) current density−voltage−luminance (J−V−L) curves of solution−process−doped OLED devices, (**d**) external quantum efficiency (EQE) vs. current density curves (inset: current efficiency vs. current density curves), (**e**) power efficiency vs. current density curves, and (**f**) EL spectra (inset: photographs of devices operating at 7 V).

**Table 1 nanomaterials-12-03806-t001:** Photophysical properties of the synthesized materials.

	Solution ^a^		Film ^b^		PLQY ^c^ (%)	*E*_S_/*E*_T_ ^d^ (eV)	Δ*E*_ST_ ^e^ (eV)	τF ^f^ (ns)	HOMO ^g^ (eV)	LUMO (eV)	*E*_g_ (eV)
	UV_max_ (nm)	PL_max_ (FWHM) (nm)	UV_max_ (nm)	PL_max_ (FWHM) (nm)							
TDBA-BCZ	317, 345, 380	413 (55)	318, 346, 383	436 (58)	34/36/37	3.28/ 2.96	0.32	3.13/7.99	−5.55	−2.47	3.08
TDBA-PCZ	333, 346, 379	451 (60)	338, 350, 382	470 (61)	48/58/52	3.02/ 2.89	0.13	5.07/12.2	−5.26	−2.27	2.99

^a^ Toluene 10^−5^ M. ^b^ Spin-coated film. ^c^ Photoluminescence quantum yield of solution, neat film, and doped film (mCP: 30% emitters). ^d^ Singlet and triplet energy level measured in toluene solution. ^e^ Δ*E*_ST_ was obtained by the difference between singlet and triplet energies. ^f^ Lifetime calculated from fluorescence decay of the neat film and doped film (mCP: 30 wt% emitters). ^g^ HOMO level was measured by ultraviolet photoelectric spectroscopy of AC-2.

**Table 2 nanomaterials-12-03806-t002:** EL performances of the solution-processed OLED devices.

EML	*V*_on_^a^ (V)	CE (cd/A) ^b^			EQE (%) ^c^			CIE (x, y) ^d^	EL_max_ (nm)	FWHM (nm)
		Max	at 1000 cd/m^2^	at 2000 cd/m^2^	Max	at 1000 cd/m^2^	at 2000 cd/m^2^			
mCP: 30 wt% TDBA-BCZ ^e^	4.89	8.67	8.26	6.85	7.73	7.34	6.16	(0.161, 0.046)	428	47
mCP: 30 wt% TDBA-PCZ ^e^	4.29	14.24	13.72	13.20	10.58	10.14	9.71	(0.151, 0.155)	461	58
mCP: 30 wt% TDBA-PCZ ^f^	8.01	1.09	1.04	-	0.32	0.30	-	(0.156, 0.232)	469	74

^a^ Turn-on voltage at 1 cd/m^2^. ^b^ Current efficiency. ^c^ External quantum efficiency. ^d^ CIE color coordinates of conventional devices at 7 V and inverted devices at 8 V. ^e^ Conventional devices. ^f^ Inverted device.

## Data Availability

Not applicable.

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
