# Peer review of "Organic–Inorganic Hybrid Device with a Novel Deep-Blue Emitter of a Donor–Acceptor Type, with ZnO Nanoparticles for Solution-Processed OLEDs"

_nanomaterials, 2022, doi:10.3390/nano12213806_

Round 1
Reviewer 1 Report
The present work by Kang et al. discussed the synthesis, characterization and device application of two blue organic emitters with an organoboron based acceptor unit and two different carbazole based donor units. They also studied the EL performance in both normal and inverted device structures. My main concern with this work is that it lacks a main focus and the novelty is unclear. Some of the descriptions seem strange for the field of OLEDs. To improve, the authors might focus on one specific topic, such as how the D-A structure affects the properties of the organoboron emitter. The contents on ZnO based inverted OLEDs need further investigations, such as improve the device performance, study the device physics, and so on, and might be better submitted separately as a new work. The manuscript in the current form is not acceptable for publication in Nanomaterials.
Reviewer 2 Report
In this manuscript, bipolar deep-blue-light-emitting materials were designed, synthesized, and characterized, targeting large-area production solution-processed OLEDs with enhanced performance.
The synthesis and characterization of organic blue emitters (e.g., photophysical properties, surface morphology, thermal properties) important for the fabrication of OLED devices are detailed. The photophysical properties of road-shaped ZnO nanoparticle-based films and the electroluminescence properties of fabricated OLED are convincing and supported by experimental evidence. Future improvement steps concerning efficiency are also pointed out.
Overall the interpretation of the results is accessible, and the obtained results logically support the conclusion. However, the device lifetime, efficiency, and narrow-band should be further discussed regarding suitability for applications.
Also, minor corrections are needed (e.g., on page 2, line 88, the reference number should be added, in the “Experimental” section, the heading number (“synthesis 2.1”) should be deleted the subsections should be corrected accordingly.
The manuscript can be published in Nanomaterials after minor revision along the above lines.
Reviewer 3 Report
Authors have presented their work on blue OLED which is very important to generate a full-color display. OLED performance depends on device design, interfacial energy band alignment, emitter design, external quantum efficiency, power efficiency, and operational stability. Utilizing ZnO nanoparticles could be beneficial in OLED technology. The author's work is impressive but requires a few additional clarifications. I suggest publication after some minor corrections and addition
1. How the molecular packing affects the PL shift in the film state compared to the solution state?
2. Authors should provide reference to the aggregation induce effect (AIE) mentioned in line 250.
3. The power efficiency plot should be included in the main manuscript.
4. How was the operational stability of the devices under constant current?
5. Do the PL properties of these emitters expected to be changed under different photon flux/power densities?
6. The lower leakage current from the L-I-V curve (Figure S7) is not quite clear. Authors are suggested to remove the data processing on the plotting or increase the number of data steps that indicate the lower leakage current. Also, please insert the LIV Curve in the main manuscript
7. Particle size distribution of ZnO with mean and standard deviation should be included for the readers.
8. A cross-sectional SEM image is advised to be included in the manuscript for both devices so that readers can see each thin film thickness and thin film stacking configuration.
Round 2
Reviewer 1 Report
As this paper will be published in a special issue (Solution-Processed Metal Oxide Nanostructures for Carrier Transport), it is better to extend ZnO related contents, such as why ZnO based device shows much lower performance than the conventional device. Otherwise, it is more suitable for journals with a focus on organic synthesis.
Round 3
Reviewer 1 Report
I have no further comments.